# Attentional Bias for Sleep-Related Words as a Function of Severity of Insomnia Symptoms

**DOI:** 10.3390/brainsci13010050

**Published:** 2022-12-27

**Authors:** Marco Fabbri, Luca Simione, Laura Catalano, Marco Mirolli, Monica Martoni

**Affiliations:** 1Department of Psychology, University of Campania Luigi Vanvitelli, Viale Ellittico 31, 81100 Caserta, Italy; 2Institute of Cognitive Science and Technologies, CNR, Via San Martino Della Battaglia 44, 00185 Rome, Italy; 3Department of Experimental, Diagnostic and Specialized Medicine, University of Bologna, 40121 Bologna, Italy

**Keywords:** insomnia, attentional bias, disengagement, visual dot-probe task, insomnia severity index, cognitive bias

## Abstract

Attentional bias to sleep-related information is thought to be a core feature for developing and/or maintaining insomnia. This study used a hallmark measure of attentional bias, the dot-probe task, to determine whether this bias toward sleep-related stimuli was a function of the severity of insomnia symptoms. A sample of 231 volunteers (175 females; mean age of 26.91 ± 8.05 years) participated in this online study, filling out the Insomnia Severity Index (ISI) and performing a visual dot-probe task. After categorizing individuals based on the ISI score into normal, subclinical, and moderate/severe sleep groups, we only found a marginally significant interaction between sleep groups and the type of stimuli on RTs, suggesting that subclinical and moderate/severe sleep groups reported slower RTs for sleep-related words than for neutral words. When we calculated the attentional bias score (ABS), we found that ABS significantly differed from zero in the moderate/severe sleep group only, suggesting a disengagement for sleep-related information as a function of the severity of insomnia symptoms. This finding seems to suggest that insomnia is related to greater difficulties in shifting away from sleep-related stimuli.

## 1. Introduction

Selective attention is commonly defined as the process by which a specific stimulus is selected by internal filtering for further processing [1]. It is worth noting that several cognitive models of insomnia have proposed that this sleep disorder is, in part, maintained by selective attention to sleep-related threat cues [2,3,4]. For example, the attention-intention-effort (AIE) pathway [3] suggests that sleep can be considered a salient stimulus. Specifically, in this model, an important role is played by the difficulty of inhibiting wakefulness (for the onset of sleep), and the development of (chronic) insomnia is determined by three related cognitive processes: attending to sleep-related stimuli, explicitly intending to sleep, and applying voluntary effort to sleep-onset. Thus, insomnia is a sleep effort syndrome characterized by excessive sleep preoccupation [2]. Related to this point, insomnia patients are selectively vigilant for stimuli indicative of wakefulness in the nighttime and symptoms of fatigue in the daytime [5]. Consequently, insomnia patients monitor their internal (e.g., muscle tension) and external environment (e.g., the clock for estimating how many hours are needed for a restorative sleep) for sleep-related threats selectively to confirm that they have not slept, and (subsequently) daily functioning is impaired [4]. Selective vigilance, in turn, can enhance worry and anxiety about disturbed sleep and its consequences, leading to an increase in autonomic arousal and emotional distress [4]. According to these theoretical accounts, attentional processes may be considered a possible psychological process that maintains chronic insomnia.

Several studies have examined the presence of a sleep-related attentional bias in insomniacs or poor sleepers. Sleep-related attentional bias is defined as the tendency to selectively attend to sleep-related information in comparison to neutral information [3]. Sleep-related attentional bias has been studied using computerized or reaction time (RT)-based experimental tasks, such as the emotional Stroop, the dot-probe, the Posner, and the flicker or change blindness [6,7,8,9,10,11,12,13,14,15,16,17,18,19,20,21,22,23,24,25,26,27], or by adopting eye-tracking paradigms [28,29,30]. The review of these studies points out mixed results with positive (i.e., presence of the attentional bias) [7,11,12,14,15,16,17,18,20,21,22,23,24,25,26,27,28,29,30] and negative (i.e., absence of the attentional bias) [6,8,9,10,12,13,19] results in insomniacs and nonclinically poor sleepers, suggesting that the sleep-related attentional bias seems to be a fragile phenomenon. In addition, mixed results have been observed in the efficacy of attentional bias modification (ABM) as a treatment for insomnia; for example, Milkins et al. [31] showed that patients with insomnia, who completed the ABM task, reported shorter sleep onset latencies and lower pre-sleep worry during the night, but, in a follow-up study [32], the authors found no effect on insomnia severity or sleep-related attentional bias. In line with these last findings, Lancee et al. [33] did not observe a change in the attentional bias or sleep problems using ABM. Furthermore, two possible hypotheses have been proposed to discuss the meaning of the sleep-related attentional bias in insomnia patients: on the one hand, patients with insomnia strongly crave getting good sleep quality [3], while, on the other hand, patients with insomnia show signs of threat and/or anxiety in response to sleep-related cues because of their long-term poor sleep quality. Although it is possible that both hypotheses can be associated with sleep-related attentional bias, it is still unknown whether the attentional responses to sleep-related cues represent a craving hypothesis or a threat hypothesis [34]. Thus, further investigation of sleep-related attentional bias is needed.

From a deep inspection of the literature review, most of the studies implemented the emotional Stroop test (9 studies) or the dot-probe task (7 studies), with great sensitivity to the dot-probe task to highlight sleep-related attentional bias in insomniacs or nonclinically poor sleepers. The difference between the two tasks, regarding the task setting and instruction, as well as which attentional component is assessed, can explain the preference for the dot-probe task in measuring sleep-related attentional bias (for a review, see [35]). In the emotional Stroop task, neutral and emotionally salient words are presented on the screen in one of four colors, and participants are required to press the response key corresponding to this specific color. In this task, it is expected longer RTs for emotionally salient words compared to neutral stimuli, as an index of Stroop interference. The emotional Stroop task is used to assess the extent to which attention toward emotional content interferes with performance when responding to non-emotional content [36]. Thus, this task elicits spatial attention and covert shifts of spatial attention to threatening stimuli, making it well-suited for studying difficulties in disengaging attention from threats [37]. In the dot-probe task, two visual stimuli (e.g., words or pictures), called cues, are briefly and simultaneously presented above and below, or to the left and right of a fixation cross. One cue is an emotional or threatening stimulus, and the other is a neutral stimulus. After the cues disappear, a probe or target (e.g., a dot) appears in the location of one of the cues. Participants must quickly and accurately respond to the location or identity of the probe. Participants demonstrate faster responses to the probe that appeared in the location of a threatening stimulus compared to a neutral stimulus, exhibiting an attentional bias toward the threat (i.e., interference index [25,27]). The dot-probe task is used to measure the attentional bias in spatial orienting to threatening cues [38] and is used to assess facilitated attention to the threat, difficulty in disengaging attention away from the threat, and attentional avoidance [37]. Thus, the dot-probe task can be considered a sort of “gold standard” in behavioral attentional bias research because the interference index indicates the interaction between word position and probe position on RTs, eliminating response bias interpretations, which are more common in the emotional Stroop task [37]. As described above, the emotional Stroop task can be a measure of increased vigilance to salient stimuli, but it also reflects the impact of heightened arousal interfering with information processing when sleep-related stimuli are presented [15,16,23]. By contrast, the dot-probe task can be considered a possible solution to this problem, and the interference index reflects a shift towards or away from sleep-related stimuli [23,37,38]. Motivated by these considerations, in the present research, we adopted a dot-probe paradigm, using words as stimuli, in an attempt to disentangle the mixed results in the literature, regarding the efficacy of these types of stimuli in detecting sleep-related attentional bias with respect to pictural stimuli [25,26,27]. Furthermore, the use of word stimuli can provide insights for explaining the mixed results reported by the studies that implemented attention bias modification training [31,32,33,39], given that all the studies that applied attention bias modification training, used word stimuli during the training.

In line with previous works [6,7,9,10,11,12,14,15,16,19,21,22,23] and with the recommendation provided by Harris et al. [25], we performed a cross-sectional study in which volunteers from the general population were requested to fill in the Insomnia Severity Index (ISI [40]) and the dot-probe task. In our opinion, the ISI is more adequate than the well-used Pittsburgh Sleep Quality Index (PSQI [41]) in detecting insomnia symptoms (for a review, see [42]). Indeed, the symptoms of and processes underlying insomnia can be considered to exist along a continuum, varying between subclinical and clinical populations in terms of severity and intensity [43]. This different characterization of insomnia symptoms can be better detected using ISI than the PSQI, and thus, the ISI represents a more valid and reliable tool to associate insomnia symptoms (severity and intensity) with sleep-related attentional bias. The use of the ISI can also identify normal sleepers, subclinical and clinical individuals, and a comparison among groups in sleep-related attentional bias, and can address the debate regarding the role of sleep-related attentional bias in maintaining insomnia [6,7] or in developing insomnia [10,15], given that it has been reported that the sleep-related attentional bias found in subclinical individuals seems to reflect a precursor of insomnia [44].

Considering the abovementioned considerations, the overall aim of this study was to examine whether individuals with different severity and intensity of insomnia symptoms (i.e., subclinical, and moderate/severe insomnia participants) showed a selective attentional bias towards sleep-related word stimuli compared to normal sleepers. Furthermore, the purpose was to investigate the nature of this potential association, hypothesizing that the selective attentional bias was characterized by vigilance (a sign of craving) for sleep-related words [25,26,27]. In other words, it was expected that the subclinical and moderate/severe insomnia groups would show faster responses to sleep-related stimuli than to neutral stimuli.

## 2. Materials and Methods

### 2.1. Participants

A sample of 231 volunteers participated in the online study, using psytoolkit [45,46]. The choice of a web-based study was based on the specific restrictions adopted by the Italian government to contrast the spread of COVID-19 (at the following link, all measures adopted by the Italian government to stop the spread of COVID-19 in reverse chronological order: https://www.governo.it/it/coronavirus-misure-del-governo (accessed on 21 March 2021). The data collection started at the end of April 2021 and ended at the end of March 2022, lasting approximately one year. In addition, an online study could guarantee a larger sample size, reducing the possibility of underestimating the statistical power [47].

Participants were unpaid, anonymous, and could withdraw from the study at any time. The link to the survey was provided to all individuals who explicitly requested it. The link was posted on major social media sites or on university campuses. Of the participants, 175 were female. The mean age was 26.91 years (SD = 8.05 years; range 18–74), and an age difference between women (M = 26.03 years; SD = 6.74 years) and men (M = 29.64 years; SD = 10.82 years) was found, with *t*(229) = 2.97, *p* < 0.005, *Cohen’s d* = 0.40. Overall, the majority (45.90%) of participants reported having a bachelor’s degree, while 29.00% had a high school degree. Another 17.30% of individuals had a master’s degree, and the remaining 7.80% of participants were equally distributed in having middle-school diploma, a university master’s degree, or a PhD. No gender difference was present in the distribution of the reported educational qualifications (*χ^2^*(5) = 8.20, *p* = 0.15). Then, we eliminated from the data analysis 10 individuals because their accuracy in the visual dot-probe task was equal to 80.50% (SD = 11.29%), that is, it was lower than the mean accuracy (M = 96.44%; SD = 1.77%) of the remaining sample (mean age of 26.80 ± 8.01 years; 76.00% of women; 45.90% of bachelor’s degrees).

The study protocol was approved by the Ethical Committee of the Department of Psychology at the University of Campania, Luigi Vanvitelli (protocol number: 16–13 April 2021), and all participants provided four different forms of informed consent prior to starting the procedure.

### 2.2. Insomnia Severity Index (ISI)

After the demographic information, all participants filled in the Insomnia Severity Index (ISI), which assesses the experience of insomnia symptoms [40,48]. The well-validated Italian version of the ISI comprises seven items and examines the severity of insomnia symptoms over the past two weeks. In addition, the ISI requested that the participants indicate the difficulty they experienced in initiating and maintaining sleep and in waking up too early. All the items are scored on a 5-point Likert scale (0 = none/very satisfied/not at all interfering/not at all noticeable/not at all; 4 = very/very dissatisfied/very much interfering/very much noticeable/very much), and the total score is calculated by summing up the scores of each item, ranging from 0 to 28. A higher total score reflects greater insomnia symptoms. According to the total score [40], the 0–7 range indicates an absence of insomnia symptoms (i.e., no clinically significant insomnia), the 8–14 range indicates subthreshold insomnia, the 15–21 range indicates the presence of moderate insomnia, and the 22–28 range indicates severe insomnia. In the present study, the Italian version of ISI reported a Cronbach’s alpha equal to 0.86, whereas inter-item correlations ranged from 0.26 to 0.64, and the item-total correlations ranged from 0.67 to 0.84 [48].

### 2.3. Visual Dot-Probe Task

To measure an attentional bias towards sleep-related cues, a dot-probe task was used and measured online, as previously performed by [33]. The dot-probe task is a computerized reaction-time task wherein two words (i.e., cues) appear simultaneously above and below a central fixation cross (+). After the cues disappear, a target probe appears at the location of one of the two cues. Participants are requested to respond as fast as possible to detect the probe [25,26,27]. In the current study, each trial began with a white fixation cross (a + in Times New Roman, font 44) displayed on a black screen for 500 ms. The fixation cross was always presented in the center of the screen (x and y coordinates: 0,0). Therefore, two white words (prepared as Photoshop files [45,46]; Times New Roman, font 36) were displayed on a black screen, either above or below the fixation point, for 500 ms. The distance from the center was 150 pixels above (x and y coordinates: 0, + 150) and 150 pixels below (x and y coordinates: 0, −150) the fixation cross. When this time elapsed, a white dot (15 pixels in diameter) appeared on a black screen, for 5000 ms or until the participant’s response, in the same spatial location as one of the word stimuli previously presented (and thus, the x and y coordinates were 0, + 150 and 0, −150, respectively). Participants were required to indicate, as quickly as possible, the location of the dot probe (above or below a central cross), using the upward or downward directional arrows of the computer keyboard. After the time elapsed or the participants provided a response, a black screen was presented for 500 ms. In a similar way to [11], we decided to present 40 neutral word-neutral word (e.g., MIRROR-BALL; Appendix B) couples (i.e., filler trials) and 40 pairs of sleep-related and neutral words (e.g., BED-WATER; Appendix B). Each couple was presented 4 times (2 spatial positions × 2 dot positions), for a total of 320 trials. Before the experimental task, all participants performed a training session with 16 different (i.e., these stimuli were not presented in the experimental session) pairs (8 sleep-related words neutral words and 8 neutral-neutral words; Appendix C), and they received feedback after their responses with the presentation at the center of the screen of the word CORRECT in green for correct detection or the word ERROR in red for incorrect detection. In addition, the word ELAPSED TIME in red could be presented if participants did not press any button. This feedback lasted for 500 ms on the screen. The task lasted approximately 30–35 min.

The stimuli were selected from the following papers [15,19,28,33,39] and translated into Italian by an English speaker (Appendix B). For both types of couples, we matched the word stimuli in each couple, in terms of number of syllables, word length, word type, and frequency of occurrence in the Italian language.

### 2.4. Procedure

As stated in Section 2.1, we first presented the study during psychological courses or via main social media. All individuals, who manifested an interest, received the link this online survey. Using psytoolkit [45,46], we set the constraint to exclusively use a laptop or a PC computer to participate in the study. Thus, tablets, mobile phones, or other electronic devices were forbidden. According to [45,46], all stimuli were calibrated with respect to the display size used to perform the study automatically, although the experimental phases were kept constant for all participants’ study [45,46].

When participants clicked on the link, a brief presentation of the study was provided with brief instructions on how to fill the ISI and perform the visual dot-probe task. After that, a text with informed consent was presented. Participants received four questions related to the reading of the informed consent, the understanding of their rights, the agreement to participate, and the consent of processing data in an anonymous form. For each question, participants could select between YES/NO or AGREE/DISAGREE options. Only individuals who selected the YES or AGREE options for all questions participated in the study. After informed consent, participants received instructions to respond to demographic information and the ISI. Then, the visual dot-probe task was presented with general instruction of the task (e.g., the explanation of this detection task using directional arrows to indicate the spatial position of the dot-probe). Then, participants received the instruction to perform the training session. After that, the instruction related to the experimental session was presented. After 160 trials, a brief break was provided. After the break, participants performed a new training session followed by a new experimental session. At the end of the attention task, a text with a brief debriefing of the study was presented together with the experiment contacts, so that participants could ask for further information if interested.

### 2.5. Data Analysis

With regards to the attentional task, we considered only correct trials. In addition, we excluded from the analyses, all RTs lower or higher than 3 SD from the mean for each participant in each possible condition (i.e., type of couple, spatial position of the word, and spatial position of the dot), considering them as outliers (4.25%). In line with [15], we further eliminated from the analyses four participants because their RTs mean was 3 SD higher than the RTs mean of the general sample.

First, we categorized participants on the basis of the ISI cut-off [40,48]. Then, we tested any gender, educational level, and age differences in these categories, using a chi-squared test and a one-way ANOVA. Thus, we inserted the variables, which provided significant results, as covariates in the subsequent analyses. For the visual dot-probe task, a three-ways mixed ANOVA on mean RTs with Sleep Group (3 levels: normal, subclinical, and moderate/severe groups) as a between-subjects factor, and with Type of Word (2 levels: sleep-related vs. neutral words), and Spatial Position (2 levels: dot above vs. dot below) as within-subjects factor, was performed considering the sleep-related-neutral words couples only. Then, we calculated the attention bias score on mean RTs using the following equation:ABS = [(SleepAProbeB + SleepBProbeA) − (SleepAProbeA + SleepBProbeB)]/2, (1)
where ABS represents the attentional bias score, sleep indicates the sleep-related words, A indicates the space above the fixation cross, and B indicates the space below the fixation cross (e.g., SleepAProbeA = mean RT when the sleep-related words and probe were both above the + symbol).

The ABS summarizes the interaction between the sleep-word position and the probe position on RTs, providing a measure of the relative speeding of RTs to probes that appear in the same position as sleep-related words. Thus, positive values of the ABS reflect vigilance for sleep-related words relative to neutral words, whereas negative ABS represents avoidance for sleep-related words. A one-way between-subjects ANOVA with the sleep group factor on the mean ABS was performed. In addition, for each sleep group, the mean ABS was tested against zero using a *t*-test. Finally, a Pearson product-moment correlation coefficient was calculated between the ABS and ISI score to establish whether the values of the ABS were associated with the severity of the insomnia symptoms.

## 3. Results

Of the 217 participants analyzed, 52.50% were categorized as normal sleepers. By contrast, 32.70% of the sample reported ISI scores for subthreshold insomnia, 13.40% for moderate insomnia, and the remaining 1.40% for severe insomnia. Given that this last category was composed of 3 individuals, we decided to insert these participants in the moderate group, creating a unique moderate/severe category, which was represented by 14.80% of the sample. Importantly, we did not find any sleep group differences (Table 1) for any of the demographic information (gender: *χ^2^* (2) = 3.17, *p* = 0.21; educational level: *χ^2^* (10) = 4.62, *p* = 0.92; age: *F*(2,214) = 0.39, *p* = 0.68). Thus, no covariates were included in the subsequent analyses.

All data are available as Appendix A. The mixed ANOVA did not reveal any significant main effect and/or interaction (all *F*s < 4.00 and *p*s > 0.05). We only observed a tendency towards the significance for the Spatial Position factor (*F* (1,214) = 3.82, *p* = 0.052, *ƞ^2^_p_* = 0.02), suggesting that the probes were faster detected when they appeared above (M = 432 ms; SD = 64.28 ms) than when they appeared below (M = 438 ms; SD = 69.06 ms) the fixation cross. In addition, the ANOVA showed that the sleep group factor (*F* (2,214) = 2.76, *p* = 0.065, *ƞ^2^_p_* = 0.03) and the sleep group x type of word interaction (*F* (2,214) = 2.51, *p* = 0.08, *ƞ^2^_p_* = 0.02) tended slightly towards significance (Table 2). In order to further investigate this tendency towards the significant for the sleep group factor, we decided to run two additional one-way between-subjects ANOVAs on general RTs and mean RTs for the neutral-neutral words couple. The ANOVA on general RTs showed a significant sleep group effect (*F* (2,214) = 3.12, *p* < 0.05, *ƞ^2^_p_* = 0.03), suggesting a general slowness in detecting the dot-probe for the moderate/severe group with respect to the other two groups, although these comparisons were not significant at post-hoc test (Table 2). The ANOVA on mean RTs for neutral-neutral word pairs revealed a significant sleep group effect (*F* (2,214) = 3.46, *p* < 0.05, *ƞ^2^_p_* = 0.03), confirming the RT pattern of the previous univariate ANOVA. Indeed, the moderate/severe group reported higher RTs than those reported by the normal group (*p* = 0.08), with the subclinical group in the middle (Table 2). Thus, the moderate/severe group was generally slower in detecting the dot probe, but this slowness remained observable when two neutral words were presented, while it slightly disappeared when neutral and sleep-related words were presented, probably due to the presence of a sleep-related attentional bias.

The sleep group x type of word interaction of the omnibus ANOVA was further investigated by the specified one-way between-subjects ANOVA on the mean ABS. This latter ANOVA confirmed that the sleep group factor tended towards the significance (*F* (2,214) = 2.51, *p* = 0.08, *ƞ^2^_p_* = 0.02). However, as displayed in Figure 1, a positive value (i.e., faster RTs for detecting probes in the same position of sleep words than those for detecting probes in the same position as neutral stimuli in sleep-neutral couples) was found for the normal group (M = +1.80; SD = 16.74), while the ABS was increasingly more negative (i.e., faster RTs for detecting probes in the same position of neutral stimuli than those for detecting probes in the same position as sleep stimuli in sleep-neutral couples) as a function of the severity of insomnia symptoms (subclinical: −2.99 ± 20.29; moderate/severe: −4.39 ± 11.62). When we tested each ABS against zero for each sleep group, no significant differences were found for normal (*t*(113) = 1.15, *p* = 0.25) and subclinical (*t*(70) = −1.24, *p* = 0.22) groups, while it significantly differed from zero for the moderate/severe group (*t*(31) = −2.14, *p* < 0.05). These findings could suggest that the moderate/severe sleep group tended to avoid the sleep-related words. However, we finally failed to find a significant correlation between the mean ABS and ISI score in the sample (*r* = −0.09, *p* = 0.18). According to [48], we also calculated the impact (items 3, 4, and 5), satisfaction (items 1a, 2, and 5) and severity (items 1a, 1b, and 1c) factors, which composed the ISI. However, we did not find any significant correlations between ISI factors and the ABS (*r* = −0.06, *p* = 0.38, *r* = −0.07, *p* = 0.29, and *r* = −0.11, *p* = 0.11 for impact, satisfaction, and severity factors, respectively).

## 4. Discussion

The aim of the present study was to examine whether individuals, reporting different severity and intensity of insomnia symptoms had a selective attention bias towards sleep-related words. Specifically, it was hypothesized that the selective attentional bias was characterized by vigilance for sleep-related words [25,26,27], and this attentional bias was associated with the severity and intensity of insomnia symptoms.

For this aim, we ran an online cross-sectional study with a large sample, which performed a visual dot-probe task with neutral-neutral words couples and neutral-sleep-related word pairs [15,17,19], and filled in the ISI [40,48]. Using the individual ISI score, we categorized all participants into three sleep groups: normal group, subclinical group (i.e., an ISI score ranging from 8 to 14, detecting subthreshold insomnia), and moderate/severe group (i.e., ISI score ≥ 15). Our results failed to show a significant attentional bias for sleep-related stimuli, especially for people who reported subthreshold or moderate/severe insomnia symptoms, in line with [12,19]. Indeed, in the omnibus ANOVA, we only found a slight tendency towards the significance for both the main sleep group effect and the interaction between the sleep group and type of words factors. The first finding suggested slower RTs in performing the task, whereas the interaction described suggested slower RTs for the moderate/severe insomnia group in detecting the probe appearing in the same spatial position as a sleep-related word compared to the other two sleep groups (Figure 1). On the one hand, the general slowness in detecting the probe could reflect a general attention impairment in individuals with insomnia [49,50]. However, we found a significant sleep group effect when we analyzed the general task RTs and the mean RTs for neutral-neutral word pairs (Table 2). Thus, the lack of a significant result when RTs for couples composed of neutral and sleep-related words were analyzed and could be associated with a specific avoidance pattern for sleep-related stimuli for subclinical and moderate/severe sleep groups, rather than a general slowness in responding when the dot-probe was presented.

In line with this speculation, the sleep group x type of words interaction was further analyzed with the ABS (i.e., attentional bias score [15,17,25]). The analysis confirmed that the subclinical and moderate/severe groups were slower in detecting the dot-probe when it appeared in the place of sleep-related stimuli with respect to when it appeared in the place of neutral stimuli, while for the normal sleep group, the opposite RT pattern was observed (Table 2). Furthermore, we reported that the negative ABS, suggesting an avoidance of sleep-related stimuli, significantly differed from zero for the moderate/severe group only, suggesting that there was an association between a selective attention bias and the severity/intensity of insomnia symptoms. This last finding could be explained by the anxiety literature, where cognitive avoidance has been suggested to result in longer response latencies due to the increased cognitive processing capacity involved in actively avoiding a stimulus [51]. In line with [16] (see also [21]), the attentional bias found in the moderate/severe group was due to disengagement from (or avoidance of) to “threatening” sleep-related words, suggesting a basic difficulty in interrupting attention from perceived threats [7,15,23]. The novel finding in this study was that the processing of sleep-related (i.e., threatening) words led to difficulties in shifting away from these threats, determining a slowdown of RTs in detecting the dot-probe. Although we found results that tended towards significance, we could advance the idea that people, reporting increasing severity levels of insomnia symptomatology, reacted with a “freezing” reaction when exposed to threatening words. This reaction, perhaps, mobilized the necessary resources and chose an appropriate strategy to tackle the threat [52]. It should also be noted that the vigilance effects in attentional bias research within the anxiety area have been observed with a short exposure time (e.g., 200 ms) of anxiety stimuli, whereas a relatively longer exposure time (500 ms) used in the current research could mask any vigilance effect, favoring instead an avoidance effect [52]. Our decision to use this exposure time was based on previous studies that used a similar visual dot-probe task with words or pictures [12,15,16,17,18,19,20,21,25]. Further studies should elucidate the role of exposure time in inducing vigilance for and avoidance of sleep-related cues. However, it is important to consider that our results could also be associated with the choice to use word stimuli. The pictorial visual dot-probe task may, more easily, induce the attentional bias. The review by [25] reported three positive results out of four studies using pictures as stimuli, and two positive results out of four (including the present one) studies using word stimuli. In addition, the effect size of attentional bias shown by pictorial stimuli was larger than that reported by word stimuli. Further studies should clarify the presence of any difference between words and pictures in the visual dot-probe task.

Although not conclusive, our data seems to support the role of attention in the AIE pathway into primary insomnia by impeding the automatic passage to sleep [2,3]. Cognitive models of insomnia have also emphasized that insomnia is, in part, maintained by an attentional bias for sleep-related “threats” cues [4,5]. Related to this point, our data seem to suggest that sleep-related attentional bias is related to the maintenance of insomnia [6,7], considering that only the negative ABS of individuals with moderate/severe insomnia symptoms was significantly different from zero. Although a negative ABS value was also present in individuals with subthreshold insomnia, suggesting that a sleep-related attentional bias may reflect a precursor of insomnia [44], this score did not differ significantly from zero. Thus, we supported the role of sleep-related attentional bias in maintaining insomnia [6,7] and not in developing insomnia [10,15], given that we found a significant negative ABS in the moderate/severe group. This assumption was also based on the similarity between our results and those found in diagnosed insomnia patients [e.g., 16]. In addition, this consideration seems to be in line with neurophysiological and functional magnetic resonance imaging studies, which usually report that patients with insomnia show hyperarousal in response to sleep-related stimuli, increased amygdala activity, and higher blood oxygen level-dependent (BOLD) activation in the precentral, prefrontal, fusiform, and posterior cingulate cortices [13,26,34,53]. Thus, it is possible to speculate that difficulties interrupting attention from a threat are more related to the maintenance of daytime aspects of insomnia and the increase in worry, anxiety, and safety behaviors [4]. However, further longitudinal studies are needed in order to disentangle this aspect, as reported by [27], because longitudinal data are absent in the literature, and heterogeneity among studies is reported [25,26,27].

The current findings could have relevance for clinical implications and future research. On the one hand, one therapeutic intervention that aims to modify attentional bias is the ABM treatment, which commonly uses the dot-probe task to retrain attention [for a review, see 54]. However, the studies that tested the efficacy of the ABM protocol for the treatment of insomnia have shown mixed results [31,32,33]. A possible explanation could be related to the fact that the ABM protocol trains attentional bias away from threatening stimuli and towards neutral stimuli, thus, reinforcing attentional avoidance of threat [54]. Thus, the present study should explain the mixed results in the literature, given that people with moderate/severe insomnia symptoms tended to avoid sleep-related words. A more prominent approach for the treatment of insomnia is represented by mindfulness-based interventions (MBIs) [55,56]. Mindfulness intentionally brings awareness to present-moment thoughts or sensations with an attitude of acceptance, patience, openness, curiosity, and kindness, which is formally practiced through a meditation focused on directing one’s attention towards the breath, body sensations, feelings, or thoughts [55,56]. Several works have highlighted the prominent role of the acceptance component of mindfulness in exerting its beneficial effects, in particular, with respect to sleep [57,58,59]: by inducing a more accepting stance, mindfulness may reduce the tendency to avoid sleep-related threat stimuli, resulting in a reduction of worry and anxiety. Future studies should investigate the potential effect of MBIs on sleep-related attentional bias.

However, there are some limitations to consider in the present study. First, we performed an online study without any control when participants performed the visual dot-probe task and filled in the ISI. Linked to this point, all participants first filled in the questionnaire and then performed the visual dot-probe task. This fixed order could introduce a priming-like effect, probably producing greater insomnia-related attention bias. Given the restrictions imposed by the COVID-19 pandemic outbreak, imposing an online study, we think that our sample was sufficiently large to reduce the impact of these limitations. Another limitation is with regards to the cross-sectional nature of the study, limiting any casual inference. Longitudinal studies could be more suitable for causal links. Related to this point, a shortcoming might be the generalizability of the findings based on an unbalanced distribution of gender with a large proportion of females. Although we did not observe any difference in gender distribution within each sleep group, we suggest taking this aspect into consideration in further studies, given that gender differences in selective attention have been proposed [60]. Finally, we did not assess sleep quality and quantity in an objective way. In addition, we categorized individuals based on the ISI score without another additional evaluation for the severity and intensity of their insomnia symptoms. Another possible improvement could come from using event-related potentials (ERPs), given that behavioral measures such as RTs are not sensitive enough to clarify the time course of hypervigilance towards or avoidance of threat-related stimuli [14,36,37].

## 5. Conclusions

In conclusion, we used a cognitive dot-probe paradigm using sleep-related words to provide evidence for a sleep-related attentional bias as a function of the severity and intensity of insomnia symptoms according to the ISI score. The present study provided weak support for the previous research performed in the field [6,7,8,9,10,11,12,13,14,15,16,17,18,19,20,21,22,23,24,25,26,27,28,29,30], with the only significant result of a negative ABS value for individuals within the moderate/severe insomnia group. This last finding provided further support for the role of attention in the AIE pathway into insomnia and cognitive models explaining the development and maintenance of insomnia [2,3,4,5]. Although our results are not conclusive, we found that individuals with moderate/severe insomnia symptoms showed an overall attentional bias (i.e., avoidance) for sleep-related words. These findings might have implications for theoretical conceptualizations and clinical interventions for chronic insomnia.

## Figures and Tables

**Figure 1 brainsci-13-00050-f001:**
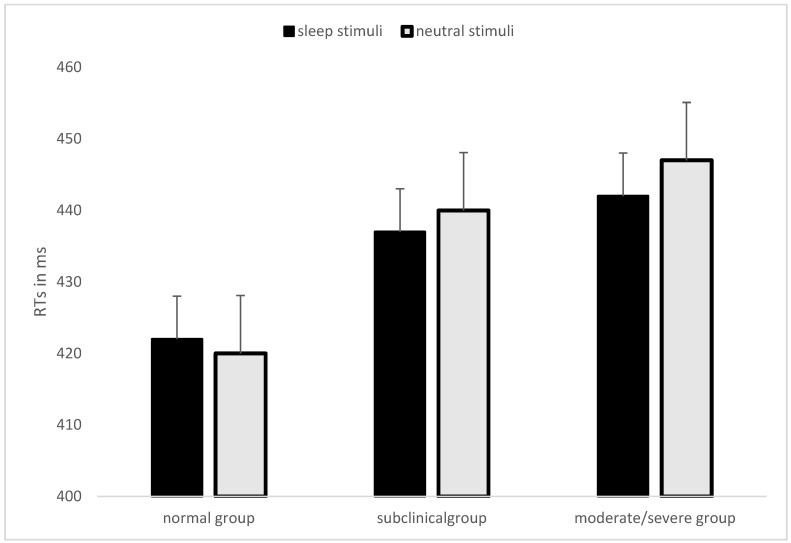
Mean RTs for detecting the probe appeared in the same position as the sleep-related stimuli (histogram in black) and mean RTs for detecting the probe appeared in the same position as the neutral stimuli (gray histogram), for sleep-neutral couples only, in each sleep group defined based on the ISI score. The bars indicate the mean standard error.

**Table 1 brainsci-13-00050-t001:** The demographic information for each sleep group created on the basis of individual ISI scores is provided.

Sleep Group	Age ± SD Years	Male	Female	Middle-School Diploma	High-School Diploma	Bachelor’s Degree	Master’s Degree	University Master	PhD
**Normal groups**	26.59 ± 7.05	53.80%	52.10%	0.90%	27.20%	49.10%	16.70%	4.40%	1.80%
**Subthreshold group**	25.72 ± 6.35	25.00%	35.20%	2.80%	31.00%	42.30%	18.30%	4.20%	1.40%
**Moderate/Severe group**	26.59 ± 7.22	21.20%	12.70%	0.00%	31.20%	43.80%	21.90%	0.00%	3.10%
**Total Sample**	26.30 ± 6.84	24.00%	76.00%	1.40%	29.00%	46.10%	18.00%	3.70%	1.80%

**Table 2 brainsci-13-00050-t002:** Mean RTs (and relative SD), expressed in ms, for detecting the probe appeared above and below the fixation cross in correspondence to neutral or sleep-related words for each sleep group are presented. In addition, the mean (and relative SD) task (general) RTs and neutral-neutral words couple RTs for each sleep group are presented. Finally, the task accuracy for each sleep group is expressed in percentage.

	Neutral Word–Probe Below	Neutral Word–Probe Above	Sleep Word–Probe Below	Sleep Word -Probe Above	Task RTs	Mean Neutral-Neutral Words Couple RTs	Task Accuracy
**Normal group** **(N = 114)**	425 (59.04)	420 (58.92)	423 (61.07)	418 (60.05)	421 (55.68)	420 (55.17)	96.54% (1.60%)
**Subclinical group (N = 71)**	441 (75.61)	433 (65.76)	440 (75.79)	440 (73.65)	439 (68.77)	439 (70.18)	96.37% (1.69%)
**Moderate/Severe group (N = 32)**	447 (69.20)	438 (61.99)	450 (73.62)	444 (65.90)	447 (65.95)	448 (67.81)	96.35% (2.22%)

## Data Availability

The excel database file is available as Appendix A.

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
