# Peer review of "Attentional Bias for Sleep-Related Words as a Function of Severity of Insomnia Symptoms"

_brainsci, 2022, doi:10.3390/brainsci13010050_

Round 1

Reviewer 1 Report

Dear Authors and Editor. The present research is simple and well constructed. The methodology is clear and simple. The theoretical framework is well constructed, although I would add more meta-analysis research and longitudinal studies to strengthen this section. On the other hand, could the objective be presented more clearly? In relation to the discussion I found it very interesting and well constructed.

Author Response

We would like to thank the Assistant Editor, Eric Yu, for the helpful comments and allowing us to review the manuscript ID brainsci-2076845. In the following part of this letter, we highlighted our responses to reviewers’ suggestions. All changes in the text were in red.

Review#1

Dear Authors and Editor. The present research is simple and well constructed. The methodology is clear and simple. The theoretical framework is well constructed, although I would add more meta-analysis research and longitudinal studies to strengthen this section”.

We would like to thank the reviewer for his/her positive comments of the manuscript. Also, we really appreciated for positive comments for methodology and theoretical framework. We quoted more meta-analysis research and longitudinal studies to strengthen this theoretical framework (lines 54 page 2). In addition, in the original version of the paper the review and meta-analysis for the theoretical framework of the paper are the following references 35, 50, 54, and 56 (the numbers correspond to the new reference order). For the longitudinal studies, as reported by recent review and meta-analysis (Akram et al., 2023, Sleep Medicine Reviews; our reference number 27) at page 12 the authors state “Given slight heterogeneity among studies and absence of longitudinal data…”. Thus, the longitudinal studies, as also we recommended at page 9 (lines 426-428) are scares. All changes in the text were made in red.

On the other hand, could the objective be presented more clearly? In relation to the discussion I found it very interesting and well constructed”.

At page 3 (lines 113-137), we tried to make the objective of the present study more clearly. We thank the reviewer for positive comment about the discussion.

Reviewer 2 Report

This is a nice and detailed study investigated the attentional bias for severity of Insomnia symptoms. The paper is structured clearly and conclusions are suported by the results. I have only one comment that expect the authors to address. Regarding the Figure 1, I suggest to use either boxplots or typical histogram to report mean ABS. In the current form, it is not a very informative graph.   

Author Response

We would like to thank the Assistant Editor, Eric Yu, for the helpful comments and allowing us to review the manuscript ID brainsci-2076845. In the following part of this letter, we highlighted our responses to reviewers’ suggestions. All changes in the text were in red.

Review#2

(x) English language and style are fine/minor spell check required

An English mother-tongue revised the paper to improve the English language and style. All changes are in red.

This is a nice and detailed study investigated the attentional bias for severity of Insomnia symptoms. The paper is structured clearly and conclusions are supported by the results”.

We thank the reviewer for her/his positive comment about the study, the structure and conclusions.

I have only one comment that expect the authors to address. Regarding the Figure 1, I suggest to use either boxplots or typical histogram to report mean ABS. In the current form, it is not a very informative graph”.

We thank the reviewer for this suggestion. Given that we reported in the text the mean and SD of ABS for each group (page 7, lines 320-326), we decided to report in the Figure 1 the mean RTs for the detection of probe in the same position of sleep (a type of “congruent” condition) and neutral (a type of “incongruent” condition) stimuli for sleep-neutral couples. We think that in this way we provided a more informative Figure.